# Children’s Health, Wellbeing and Academic Outcomes over the Summer Holidays: A Scoping Review

**DOI:** 10.3390/children11030287

**Published:** 2024-02-27

**Authors:** Emily Eglitis, Aaron Miatke, Rosa Virgara, Amanda Machell, Timothy Olds, Mandy Richardson, Carol Maher

**Affiliations:** 1Allied Health and Human Performance and the Alliance for Research in Exercise, Nutrition and Activity (ARENA), University of South Australia, Adelaide, SA 5001, Australia; emily.eglitis@mymail.unisa.edu.au (E.E.); aaron.miatke@mymail.unisa.edu.au (A.M.); rosa.virgara@unisa.edu.au (R.V.); amanda.watson@unisa.edu.au (A.M.); timothy.olds@unisa.edu.au (T.O.); 2Centre for Adolescent Health, Murdoch Children’s Research Institute, Parkville, VIC 3052, Australia; 3Office for the Early Years, Department for Education, Government of South Australia, Adelaide, SA 5000, Australia

**Keywords:** child health, physical activity, obesity, time use, summer, holidays

## Abstract

Background: The school day provides a supportive and stimulating environment that may protect children and adolescents (5–18 years) from behaviours that are adverse for health and wellbeing. Objective: To review the literature regarding changes in children’s academic achievement or overall wellbeing during the extended school summer break and evaluate if the outcomes are different for children experiencing disadvantage. Methods: The peer-reviewed literature was searched across six electronic databases for studies tracking changes in any academic, health or wellbeing outcome in children over the summer holidays. Studies were screened in duplicate for inclusion. Data were extracted using a standardized data extraction form. Outcomes were coded as decline (suggestive or significant), increase (suggestive or significant) or mixed/neutral and then compared to the school year or according to disadvantaged. Results: Seventy-six studies (*n* = 14,230,846 participants) were included. Strong evidence was found of a decline in academic outcomes and increases in adiposity, sedentary behaviour and screen time. There was moderate evidence of declines in cardiovascular fitness and physical activity. These patterns were magnified for disadvantaged children. Limited data were available on muscular fitness, sleep, diet quality and social, emotional or mental wellbeing. A total of 80% of studies were from the United States. Most data were from children 12 years of age and younger. Conclusions: Over the summer break, children’s academic and health outcomes decline. Children experiencing disadvantage display magnified losses that warrant further investigation. The summer holidays present an opportunity to improve children’s health and wellbeing.

## 1. Introduction

In addition to academic achievement, school plays an important role in children’s health and wellbeing. Schools often have a positive influence on children’s emotional, social and psychological development [1,2] as wellbeing is supported through engagement with teachers, peers and the school environment [2]. Furthermore, the school day is governed by curricula and policies regarding physical activity (PA) and diet, which enable children to display healthier patterns of behaviours relative to non-school days such as weekends and holidays [3,4,5]. Therefore, the absence of school (e.g., on weekends and holidays) has the potential to impact a wide range of health and wellbeing outcomes.

Summer holidays are the longest break from school and can last from six weeks (across Asia and Australia) to over three months in some parts of Europe and the US [6]. The education literature has long reported the phenomenon of the “summer slide”, where students experience a decline in academic achievement during the extended summer break [7]. Emerging evidence suggests that children’s obesogenic behaviours intensify on holidays and weekends, leading to weight gain [5,8], with evidence that the prevalence of childhood obesity increases exclusively in the summer [9,10,11]. Furthermore, evidence suggests that for some children, losing the connection and social support of school leads to feelings of loneliness and anxiety over the summer holidays, detrimentally affecting mental health [12].

Children disadvantaged by race, low income or social status may be affected to a greater extent. Outside of the school environment, low-income families face specific barriers to providing safe and stimulating care for children, resulting in different, and less enriching, holiday experiences for disadvantaged children [12,13]. Disadvantaged children are therefore particularly susceptible to worsening mental health over the summer [14], but the extent of these challenges remains unclear.

Cooper and colleagues’ seminal paper [7] was the first to comprehensively review the literature regarding summer learning loss over a quarter century ago and demonstrated a summer decline in academic achievement that disproportionately affected disadvantaged children. Since then, it has been proposed that the accumulating effects of multiple summer holidays lead to a widening of the academic achievement gaps between the rich and poor [15]. Currently, there is debate whether inequality in achievement scores increases [16], remains stable or decreases [17] when school is not in session.

Previous research has mainly examined children’s weight gain or fitness decline over the summer without adopting a holistic lens [5,9,10,11,18]. A few narrative reviews have described weight and fitness changes during the summer, emphasising attempts to address these issues [5,10,18]. These reviews underscored uncertainties about the most vulnerable populations, with some evidence suggesting disadvantaged children bear a disproportionate burden. These disparities hint at summer breaks exacerbating health inequalities between socio-economic groups and the existing literature reviews seem to overlook children’s social, emotional, or mental wellbeing during the summer.

In this context, our objective was to holistically evaluate current evidence across multiple domains during the summer. We hope this review will guide future research trajectories and highlight pivotal intervention areas. We conducted a scoping review to answer the following questions regarding changes in children’s health, wellbeing and academic achievement over the summer holiday period and the associations with socio-economic disadvantage: What are the geographical and historical trends in the literature related to children’s academic, health and wellbeing outcomes over the summer?How do children’s academic, health and wellbeing outcomes change over the summer holiday period?Do these changes differ according to the level of disadvantage in children?

## 2. Methods

A review protocol was developed a priori using the framework of the Joanna Briggs Institute [19], informed by the Synthesis Without Meta-analysis (SWiM) guidelines [20], and prospectively registered with Open Science Framework [21]. The review is reported in accordance with the PRISMA-ScR guidelines [22].

### 2.1. Eligibility Criteria

This scoping review aimed at providing a broad overview of the existing literature. The PCC (Population, Concept, Context) framework was used to structure the inclusion criteria: studies were considered if they focused on school-aged children aged 5 to 18 years (Population), covered any aspect of health, wellbeing, or academic outcomes (Concept) and pertained specifically to the summer holiday period, aiming to delineate changes over the summer or differences relative to the school year (Context). The research designs either tracked changes over the summer period through longitudinal data or provided comparative measures between the summer holidays and the school year. All the included studies had a minimum of 100 participants and were published in English between the year 2000 and 5 September 2022. Studies were excluded if they were from the grey literature, failed to meet the inclusion criteria or were not published in English (a detailed description of the review’s inclusion criteria is presented in Appendix A). 

To ensure a multi-disciplinary approach, we performed a simultaneous search across six major academic databases: Medline, PsychInfo, Embase, JBI (OVID), ERIC (ProQuest) and Scopus, with the searches completed on 5 September 2022. The search strategy for this review was designed to be highly comprehensive, given the study’s aim to provide an extensive overview across a broad range of the literature pertaining to education (academic outcomes), health (physical health and health behaviours) and psychology (social, emotional and mental wellbeing). The strategy was developed in close consultation with an academic librarian and refined by preliminary searches. Subject headings, keywords and MeSH using the population descriptions “children” and “adolescent” were combined using the operator AND with keywords describing the time period “summer” adjacent to the term “holidays” or “vacation”. To maintain the review’s focus, we systematically excluded studies related to COVID-19 by employing the NOT operator with the terms “pandemic”, “COVID-19”, and “SARS-CoV-2” (a detailed example of the Medline search strategy has been provided in Appendix A). 

References were managed and duplicates removed using Endnote (Endnote 20, Clarivate, Philadelphia, PA, USA). To determine the eligibility of studies for this review, title and abstract screening was followed by full-text screening: both rounds were conducted in duplicate by the main author (EE) and a second author (AM, AW, RV) using Covidence software (Veritas Health Innovation, Melbourne, Australia, available at www.covidence.org) and disagreements were resolved by consensus (by both authors). For the sixty-nine studies that met the inclusion criteria, the reference lists were searched (by EE) for further relevant studies and sixteen further studies identified. 

### 2.2. Data Extraction

Data extraction was completed using Covidence by the main author (EE) with a charting form, informed by the Joanna Briggs Institute [19], developed collaboratively by the entire authorship team and rigorously piloted prior to use. Data extraction was completed in duplicate for 20% of the studies, with discrepancies solved by consensus before the completion of data extraction by the main author, who discussed uncertainties with another author. The data extraction fields included study characteristics (author, year, country of origin, sample size), participants’ demographics (age, sex, measures of disadvantage) and data items for academic outcomes (e.g., reading, mathematics), physical health (e.g., body composition, fitness), health behaviours (e.g., PA, sedentary behaviour, diet, sleep, screen time), mental health (e.g., anxiety, depression) or social-emotional function (e.g., antisocial behaviour). During extraction, changes in these outcomes were coded in terms of the direction of change (increase or decrease) and amount of change (significant or suggestive) in the following manner: change over the summer was coded as significant if it reached the study’s own parameters of statistical significance, or no change if it did not. Changes not tested for significance were categorized as suggestive. 

### 2.3. Certainty of Evidence

The certainty-of-evidence grading was adapted from previous scoping reviews [23] based on the consistency of the findings when at least five studies were available. When at least 75% of the findings were consistently in the same direction, the evidence was graded as “Strong”. A “Moderate” grade was assigned when most of the studies (but <75%) found a pattern in the same direction. When approximately equal studies indicated a change in conflicting directions, a “Mixed” grade was applied [23]. Outcomes with insufficient data (<5 studies) were designated as “Limited” grade.

### 2.4. Synthesis of Results

The data synthesis adhered to the PRISMA-ScR extension [22] and the SWiM guidelines [20]. Participant age was categorized based on standard educational groups: lower primary (ages 5–7), middle primary (ages 8–10), upper primary/middle school (ages 11–13), and high school (ages 14+). Studies were assigned to these categories based on the predominant age of the participants, or to more than one category when the distribution was evenly spread across multiple categories. 

Socio-economic status (SES), either direct or by proxy, was the primary determinant of disadvantage used in this review, while other secondary markers (e.g., race/ethnicity) were used in the absence of SES data. The SES measures included individual/family metrics (e.g., parental education, household income) or broader metrics such as school location or eligibility for school meal subsidies. 

The outcomes were primarily sorted into four domains: academic, physical health, health behaviours and emotional–social and mental wellbeing combined. Within these groups, finer analyses were conducted where uniformity in the outcomes allowed. A vote-counting method was adopted to integrate the findings. Changes were coded as *better* or *worse* over the summer. When a study reported multiple significant and suggestive changes in a consistent direction, the more robust value was retained. Studies with conflicting findings (e.g., opposite findings for subgroups with no overall measure) were counted as mixed/no difference. The impact of the summer holidays on disadvantaged children, when presented, was specifically delineated.

## 3. Results

A total of 11,870 studies were identified, with 3529 duplicates removed before the title and abstract screening (*n =* 8341). In situations where multiple reports emerged from a single study due to varied analyses or additional years, these were considered duplicates to prevent oversampling bias and consolidated into one study for analysis. Consequently, 76 studies, derived from 85 reports, were included The evidence selection sources are presented in the PRISMA-ScR flow diagram (Figure 1).

### 3.1. Characteristics of Included Studies

The studies included in this review used either primary or secondary data sources, examples of which included the Early Childhood Longitudinal Study, the Northwest Evaluation Association’s Growth Research Database, the Baltimore Beginning School Study, or administrative records. Out of these, 40 studies collected data for a single summer, 34 spanned multiple summers and 2 studies contrasted behaviours in the summer with the school term. A total of 68% (51/76) compared summer changes with school year changes. The characteristics of included studies are detailed in Table 1.

A total of 90 outcomes were reported in the 76 included studies. Academic outcomes were the most frequent, appearing in 55% (*n =* 42) of studies. Physical health was covered in 35% (*n =* 27), while 20% (*n =* 15) discussed other health behaviours. Due to the small number of studies (*n =* 4: 5%), social, emotional or mental wellbeing were grouped together. All the studies encompassed both genders, and there was considerable overlap in the age categories, with 34% (*n =* 26) of studies covering multiple age groups. Most research targeted children under the age of 12 years, from the lower primary (*n =* 41 studies: 54%, from school entry to second grade: ages 5–7 years) and middle primary age groups (*n =* 40 studies: 53% in grades three through five: ages 8–10 years). Middle school students were the subjects in 27 (36%) studies, while high school students were under-represented as participants in only 4 studies (5%). Ages were not reported in three studies (4%). Twenty-nine studies (38%) stratified results for disadvantaged populations, based on either socio-economic status or race/ethnicity.

### 3.2. Geographical and Temporal Trends in Research

Figure 2 displays the temporal trends in the research. Overall, there has been a steady increase in the volume of studies conducted across 12 years (2000 to 2022). Out of the total of 76 studies, 80% (*n =* 61) originated from the United States. Europe contributed 10 studies (13%): Austria and the United Kingdom (*n =* 2 each), Portugal, Hungary, Cyprus, Greece, the Netherlands and Germany (*n =* 1 each). The remaining 4 studies (5%) were from the Asia-Pacific region (Japan *n =* 2, New Zealand *n =* 2). Throughout the years, academic outcomes remained the primary focus. Physical health outcomes began gaining traction in the year 2006, and health behaviours started becoming prominent around the year 2013.

### 3.3. Academic Outcomes over Summer

The academic outcomes explored included literacy (*n =* 25 studies; encompassing reading ability, comprehension, vocabulary, and language); numeracy (*n =* 7 studies; covering math concepts, problem solving, reasoning); and others, including combined achievement tests, cognition (focusing on working memory) or academic progress (*n =* 14). Forty-six academic outcomes were reported from twenty-seven studies (some studies reported multiple outcomes) and are presented in Figure 3.

Overall, 41% (19/46) of studies examining academic outcomes showed a summertime decline in academic performance [29,42,60,66,67,71,79], while 37% (17/46) showed mixed findings [28,56,57,65,70,78,83,84,89,90], and 22% (10/46) indicated improvements [16,24,34,51,53,58,59,105]. A total of 89% of studies (24/27) comparing academic performance during the summer vs. the school year predominantly revealed a dip during the summer (not shown in Figure 3) [16,24,25,27,28,29,38,40,44,48,49,56,62,63,66,67,70,71,82,90,91,97,98,103,106]. The numeracy and literacy results varied. The numeracy outcomes frequently reflected a summer decline, whereas the literacy outcomes were more evenly distributed between decline, mixed findings, and improvement. Overall certainty of evidence: Strong evidence of a summer decline in the numeracy and combined measures; mixed evidence regarding literacy.

### 3.4. Physical Health

Twenty-five studies evaluated changes in adiposity over the summer using measures of body size, composition or weight status. Fourteen studies tracked changes in fitness over the summer, primarily assessing cardiovascular fitness (CVF) (*n* = 12). Studies also measured muscular strength/endurance [32,43,49] or flexibility [26,37,43] (n = 3), agility [37,43] (*n* = 2), balance [43], or motor control [33] (*n* = 1 each). 

Of the studies measuring adiposity, 84% (21/25) found an increase over the summer [9,31,32,35,37,43,45,47,49,52,54,61,64,73,74,75,76,85,86,93,96,99,100,101]. Two studies (8%, 2/25) displayed mixed results [26,95] and two more showed a decrease in adiposity over the summer [39,104] (Figure 3). To explore these increases within the context of children’s growing bodies, further analysis was undertaken. Where data were available, comparisons were made between the rates of change in the summertime compared to the rest of the school year. The pattern remained: Of the 22 studies that measured changes in adiposity across both the summer and school periods, 14 studies (64%) found comparatively greater rates of weight gain over the summer [9,35,36,37,43,47,54,64,73,74,75,76,85,92,93,96,99,100,101], seven (32%) found no difference [26,31,45,52,61,86,104] and one (4%) found smaller rates of gain [39]. Regarding CVF, the results were split between declines and mixed results, with five studies (45%) each showing declines [26,31,49,52,80] or mixed/neutral results [17,32,37,81,86,93] and only one study demonstrating an improvement [43]. These findings were further explored within the full-year pattern (not shown in Figure 4): Ten studies compared CVF changes across the summer to changes over the school year (not shown in Figure 4): Seven studies (78%) found CVF was worse over the summer [26,31,37,52,80,86,99,100,101], one Hungarian study found no difference [93] and one Austrian study found summer improvements [43]. Regarding muscular fitness, one study showed a summertime decrease [32] and two an increase [43,49]. All the flexibility results (2/2) were neutral [26,37,43]. Overall certainty of evidence: Strong evidence that children’s adiposity increases over the summer holidays. Moderate evidence of a decrease in CVF over the summer, especially when compared to the school year; evidence regarding changes in muscular fitness or flexibility is limited. The evidence suggests a decline in children’s physical health during the summer holidays. 

### 3.5. Health Behaviours

Ten studies reported on summertime changes in health behaviours, primarily related to PA. The outcomes included moderate-to-vigorous physical activity (MVPA) (*n* = 8), light physical activity (LPA) (*n* = 4), sedentary behaviour (SB) (*n* = 7) (all shown in Figure 5), screen time (*n* = 5), sleep (*n* = 3) or diet quality (*n* = 3) (all shown in Figure 6). 

The findings showed a trend towards decreasing PA over the summer. Five (63%) studies reported declines in MVPA [55,75,88,94,95,96] and n = 3 (75%) reported reduced LPA [75,94,95,100]. Conversely, n = 3 (38%) and n = 1 (25%) studies reported increases in MVPA and LPA, respectively. Sedentary behaviour and screen time consistently increased over the summer, with increases demonstrated in n = 5 (71%) of SB [69,75,94,95,96,100] and 80% (4/5) of screen time studies [50,55,88,100]. One study each showed no change [88] or a decrease [77] in SB. Only one Japanese study showed a decrease in screen time [95].

Two studies (2/3, 67%) measuring sleep found an increased duration over the summer holidays [55,100] and one found a decrease [100]. Diet quality was demonstrably worse in two studies (2/3, 67%) [96,100] and mixed in another [93] (Figure 6). Overall certainty of evidence: Strong evidence that SB and screen time increase in the summer; moderate evidence that PA declines (MVPA and LPA). Evidence regarding changes in sleep and diet quality is limited.

### 3.6. Mental, Emotional, and Social Wellbeing

Four studies reported on emotional and social outcomes, with no studies addressing mental health outcomes. Rulison and colleagues examined the relationship between adjustment and peer group aggression and reported that victimization increased in the summer relative to the school year, with the aggression level of the affiliated peer group highlighted as an important factor [87]. Light et al. focused on antisocial actions and experiences of social victimization and found an increase in antisocial behaviour over the summer, with a decline at the start of the school year [68]. Downey et al. considered how the summer period affected existing disparities in mental wellbeing between advantaged and disadvantaged groups and pinpointed persistent gaps between high and low socio-economic status (SES) groups for behaviours including self-control and interpersonal skills. These disparities, evident from kindergarten onset and evident through 2nd grade, did not seem to change over the summers [41]. Sallis et al. measured changes in self-efficacy alongside changes in PA over the summer holidays [88]. The reduction in MVPA by 14 min/day did not correlate with children’s self-efficacy, which remained unchanged over the summer. Decreased social interaction was associated with less enjoyment of PA and influenced time spent on PA. Overall certainty of evidence: Limited. 

### 3.7. Summertime Changes and Disadvantage

Twenty-nine studies presented stratified results for disadvantaged populations, based on either socio-economic status or race/ethnicity. Overall, the results showed similar patterns of summertime decline as shown for the general child population, but the patterns of loss were relatively greater for disadvantaged children. For example, amongst the studies that measured the academic outcomes of both disadvantaged and advantaged children and presented stratified results, there was a strong pattern of summertime academic loss in literacy for disadvantaged children that was not evident in the general population and comparative achievement across all the outcomes was consistently worse for disadvantaged children (Figure 7). A summary of the findings related to disadvantage across all the outcomes is presented in Appendix A.

## 4. Discussion

This review of children’s health and wellbeing aimed to synthesize the current evidence regarding summertime changes across a wide variety of domains. We explored geographical and historical trends in the research, described how children’s academic, health and wellbeing outcomes changed over the summer holidays and examined trends for children experiencing disadvantage. The findings revealed a historical focus on academic outcomes, with data being predominantly derived from US studies. In the past decade, studies from outside the US have increased, with growing attention focused on health and health behaviour outcomes. Our review highlighted summertime numeracy declines for all children, while disadvantaged children also experience declines in literacy. Children’s physical health is worse across the summer holidays, with evidence of declines in fitness and increases in adiposity compared to the school year. Clear patterns emerged of increased sedentary and screen time in the summer holidays and health behaviours are relatively worse for disadvantaged children. There was a lack of data regarding children’s social, emotional and mental wellbeing across the summer holidays.

The decline in numeracy demonstrated here is congruent with findings from two recent review studies of academic outcomes over the summer (both published in 2023), which similarly found a decline in numeracy and mixed literacy results [107,108]. Numeracy is commonly measured using standardized tests of mathematics, which have been criticized for the emphasis on procedural knowledge over deeper conceptual knowledge [109]. Procedural knowledge (e.g., knowing how to perform problem-solving steps in the correct order) relies on memorization and repetition and is therefore more susceptible to declines from gaps in practice. Children may practice procedural literacy skills in the summer through independent reading and journaling (and even interactive video games), whereas math may be less integrated into the daily routine [110]. 

Children’s physical health is better during the school year compared to the summer holidays. The summertime increases in fatness and decreases in fitness displayed here have been previously observed in the US and internationally, with the magnitude of effects demonstrably greater in regions where the summer holidays are longer [31,49,92,100,111] compared to where the breaks are shorter [112]. Left unaddressed, these summertime declines in physical health could accumulate. This is concerning because childhood obesity is a major, global public health concern with life-long consequences [113] linking it with a variety of health problems, including type 2 diabetes, heart disease and depression [114]. Physical fitness is considered one of the most important markers of health and a predictor of illness [115], and low CVF in childhood is related to increased adiposity and cardiovascular disease risk factors, poorer mental health and lower academic achievement [115]. Although the aetiology of summer weight gain is not clear, the physiology of weight gain is well established to be related to a positive energy balance in summer due to changes in key behaviours: Diet, PA, sedentary behaviour, screen time and sleep [5,10,18].

Sedentary behaviour and screen time were clearly greater during the summer. This is consistent with the Structured Days Hypothesis [3], which posits that on school days, which are purposefully planned and supervised by adults, it is easier for children to display healthier patterns of behaviour than it is on unstructured days like weekends and holidays. More sedentary behaviour means less energy expenditure over the summer and, although data on diet in summer were missing here, there is recent evidence that children tend to display worse dietary habits over the summer holidays [112] and when engaged in sedentary behaviours (e.g., snacking on energy dense, high-fat, high-sugar food and drinks), which may further tip the energy balance towards weight gain [116].

Our review showed mixed results for PA, hinting at unexplored nuances within specific subgroups (e.g., gender, age, weight status [18]). Factors like urban vs. rural settings [5] and socio-economic status can influence holiday activities [18], and detailed data on participation in organized sports or summer programs were often not tracked in the included studies.

Our review extends the current evidence base by exploring trends in each outcome for disadvantaged children. Regarding academic outcomes, disadvantaged children experienced overall sharper declines and decreases in literacy that were not apparent in the general population. These trends are concerning because summer holiday declines can accumulate and influence future curriculum pathways and career opportunities, which would limit opportunities for upward mobility through socio-economic classes [24]. Entwisle and colleagues [117] proposed the “faucet theory” to explain the comparatively worse academic outcomes over the summer for disadvantaged children. They proposed that school turns on a resource faucet for all children, which essentially turns off over the summer. While middle- to high-income families have the resources to provide enriching experiences that replace school’s influence, poorer families do not. The concept behind the faucet theory may also apply to health behaviours, which then impact physical health. 

The family environment is an important determinant of children’s health and wellbeing. All health behaviours were worse for disadvantaged children (noting that only the US studies made these comparisons). Low-income families face cost, childcare and emotional pressures over the summer holidays [118]. Difficulty finding affordable childcare results in children spending time unsupervised [119], and when left to their own devices, children self-select less healthy snacks [120]. Consecutive summer days in these circumstances influence children’s physical health. The observed differences in time use between social classes over the summer have led Weaver and colleagues [5] to propose the Health Gap Hypothesis, whereby families in the middle/upper income brackets display fewer obesogenic behaviours over the summer than children from low-income backgrounds. As a result, disadvantaged children gain weight and lose fitness at a faster rate, which increases the health disparity between the rich and poor.

This review has important strengths. This is the first ever review to bring together diverse academic, health and wellbeing outcomes in a single paper. A comprehensive search strategy was employed to cover a wide-ranging literature base, spanning the academic, health and psychology domains, enabling a complete picture to be painted of these changes in children across the summer. This review also explored trends related to socioeconomic and sociodemographic disadvantage, which helps to highlight when exacerbation of health and achievement disparities could occur.

Most of the study’s limitations arose from the literature base itself. Over three quarters of the studies were from the United States, with limited data from other world regions (particularly the Asia-Pacific and Africa). Our review did not explore differences according to age, which may be an important factor in summer holiday health [10]. Given the sparse representation of high-school participants, applying these findings to older adolescents warrants caution. In practice, there is difficulty measuring the summer period, as children are more accessible to researchers during the school period, but inclusion of in-term data can mask some summer effects. Other considerations pertain specifically to measuring changes in children’s body weight over time, as it can be challenging to delineate normal healthy growth from unhealthy or excessive increases in body weight. To address this concern regarding adiposity and CVF, where the studies provided further measures (i.e., a third time point) we compared the changes in summer to the school period and found the same consistent pattern of health declines. Finally, given the breadth of this scoping review, it was not feasible to search and include grey literature sources; however, the reference lists of included studies were searched and the authors contacted to identify other relevant studies, which is considered an effective and suitable method of addressing publication bias [121]. 

## 5. Implications

The pause or decline in learning rates experienced by disadvantaged children seems to accumulate, contributing a great deal to overall achievement inequality [122]. Despite extensive public health efforts, the incidence of childhood obesity continues to rise and differentially effect certain socio-economic groups [123]. While health interventions delivered in schools have been convenient, it seems some of the benefits gained are offset by the deterioration over the summer [5,124,125]. In this context, our review suggests that summertime interventions could improve children’s academic and health outcomes. 

Furthermore, the different summer experiences between the rich and poor may underlie the greater academic and health losses evident for disadvantaged children revealed in this review. Disadvantaged families face challenges in providing safe and enriching care over the summer [118], which could influence the amount of obesogenic behaviours these children display. Structured summer programming is one potential solution to help children and families deal with the challenges of the summer break. Summer programs would provide both childcare for carers (thus providing the primary caregiver(s) with respite and the ability to continue paid employment) while providing a stimulating environment to promote healthy behaviours in children and offer one potential solution.

Summer programs already exist globally, in a variety or structures and formats. For example, summer camps are common in the US but are perhaps under-utilized in other parts of the world. Typical US summer camps follow either an overnight format (where children spend multiple nights away from home) or day camps whereby activities and meals are provided during the day and children return home each evening. Such camps are run by a range of different stakeholders, including private, religious and not-for-profit groups, and cater to various populations (e.g., general, low-income, special needs). Alternatively, in Belgium, local councils collaborate with residents to run free “Play Streets” that provide a free space and activities for children to play with and interact with others [126]. Understanding the key elements of such programs that make them effective (and if the effects are different for disadvantaged children) remains an area for further exploration. Despite the demand for such programs, cost is a common and significant barrier to access for low-income families [127]. Therefore, the development and successful implementation of summer programs will require policy makers, providers and stakeholders (i.e., local governments, schools, children and parents) to design programs that cater to the needs of their communities and give special consideration to making programs acceptable and feasible to the most at-risk populations.

This review has highlighted gaps in the literature that warrant further research. First, a deeper exploration of the role of disadvantage across all the health and wellbeing outcomes is needed. Second, this review revealed a dearth of research into changes in children’s mental, emotional, and social wellbeing over the summer. Further research is needed to understand the broader wellbeing impact summer has on children’s mental health and if, as demonstrated for the health and academic outcomes, the trends for disadvantaged children are worse. By understanding the full scope of the challenges that children face in the summertime, comprehensive interventions can be developed.

## 6. Conclusions

This review synthesized the current evidence regarding the summertime changes in children’s academic, health and wellbeing outcomes and the associations with disadvantage. Academic outcomes stall or decline and patterns are stronger for disadvantaged children. There are unhealthy changes in adiposity and fitness over the summer, along with increases in the time spent sedentary and on screens. Data on children’s social, emotional and mental wellbeing changes across summer are lacking. The findings suggest that summertime is contributing to children’s health inequities; therefore, interventions delivered over the summer could be a promising step towards solving complex problems such as childhood obesity and the health and achievement disparity between the rich and poor.

## Figures and Tables

**Figure 1 children-11-00287-f001:**
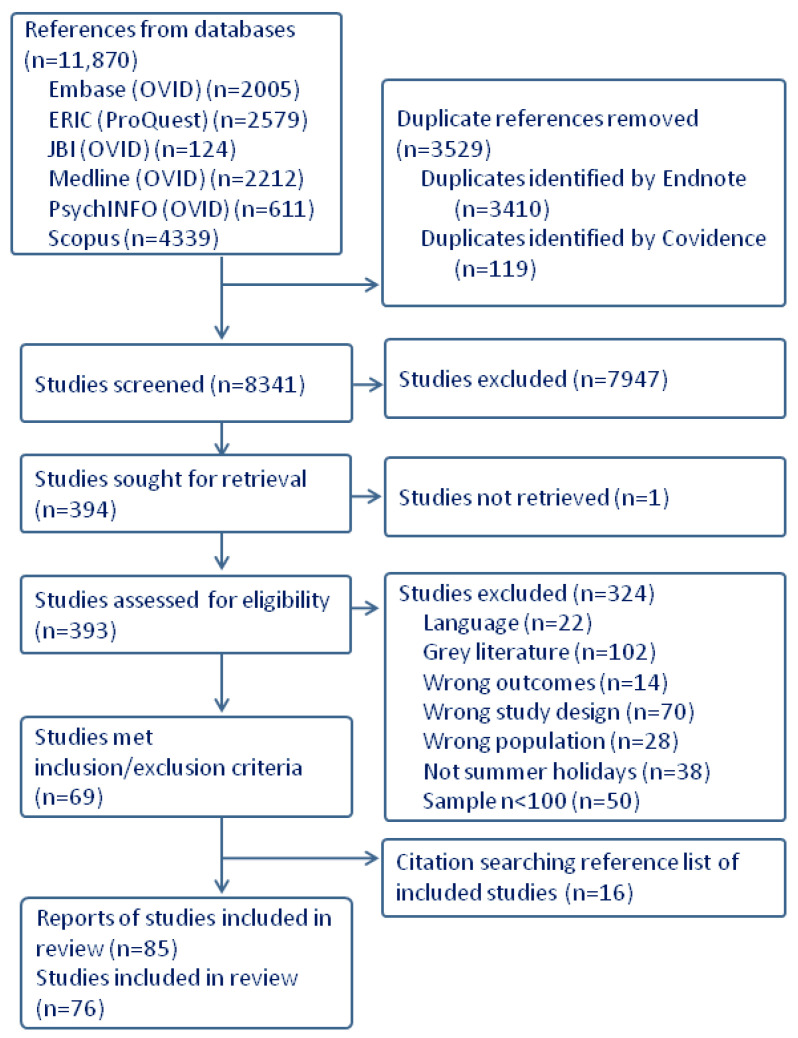
PRISMA diagram of included studies.

**Figure 2 children-11-00287-f002:**
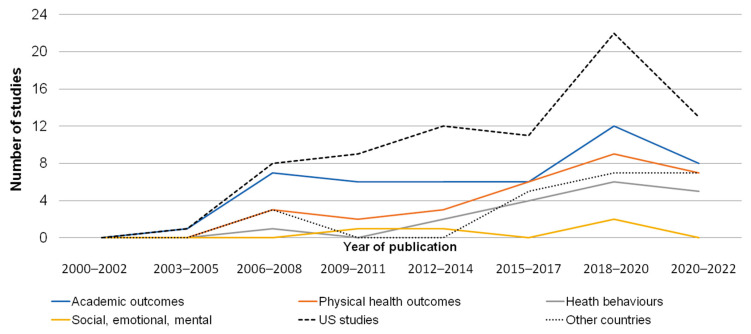
Temporal trends in research on children’s summertime academic outcomes, health, and wellbeing. Number of studies published for each outcome type and geographic location (US-based or outside the US) between 2000 and 2022.

**Figure 3 children-11-00287-f003:**
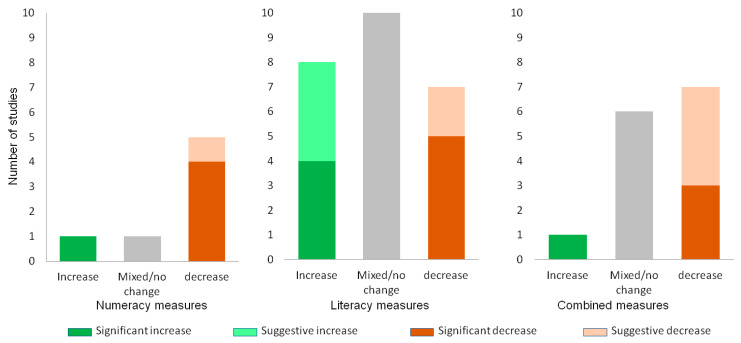
Academic outcomes over the summer. Outcomes presented according to the measure used (numeracy, literacy or combination). Statistically significant changes are shaded in a dark colour. Suggestive changes (not testing/reaching significance) are shaded in a light colour. Non-significant or significant and conflicting changes are shaded grey.

**Figure 4 children-11-00287-f004:**
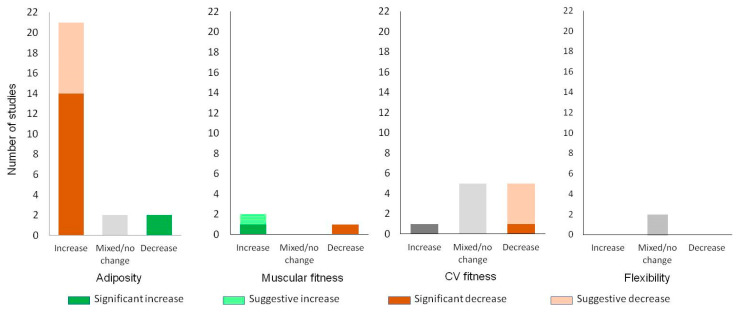
Physical health changes over the summer. Physical health changes using measures of adiposity and fitness. Statistically significant changes are shaded in a dark colour. Suggestive changes (not testing/reaching significance) are shaded in a light colour. Non-significant or significant and conflicting changes are shaded grey.

**Figure 5 children-11-00287-f005:**
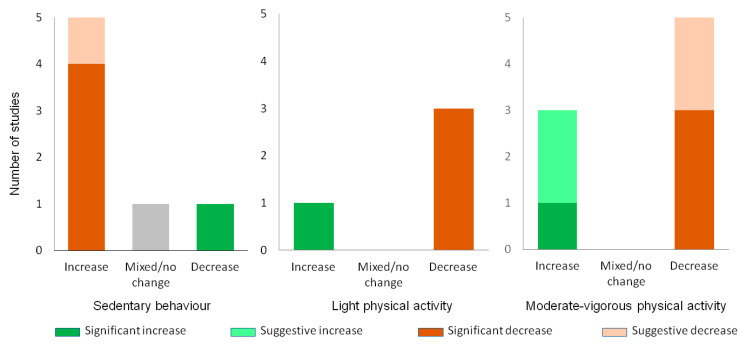
Physical activity changes over the summer. Changes in MVPA, LPA and SB across the summer. Statistically significant changes are shaded in a dark colour. Suggestive changes (not testing/reaching significance) are shaded in a light colour. Non-significant or significant and conflicting changes are shaded grey.

**Figure 6 children-11-00287-f006:**
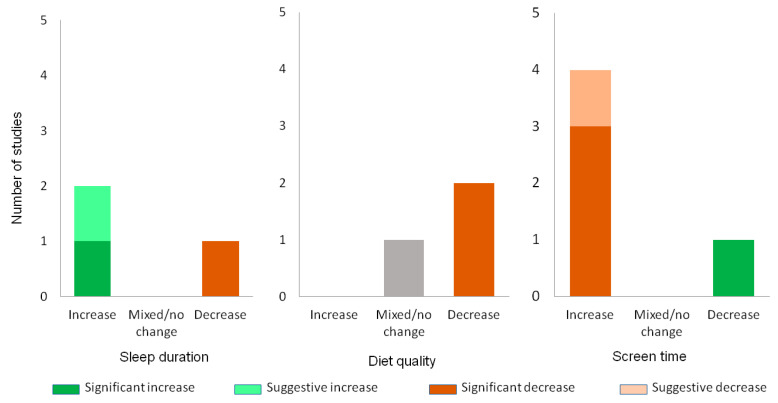
Changes in other health behaviours over the summer. Changes in sleep, diet and screen time behaviours across summer. Statistically significant changes are shaded in a dark colour. Suggestive changes (not testing/reaching significance) are shaded in a light colour. Non-significant or significant and conflicting changes are shaded grey.

**Figure 7 children-11-00287-f007:**
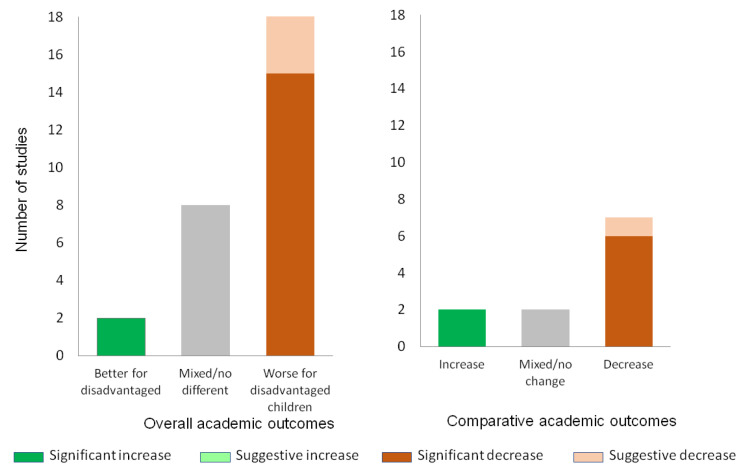
Academic outcomes of disadvantaged children over the summer. Academic outcomes for disadvantaged children (**left**) and academic outcomes for disadvantaged children comparatively (**right**). Statistically significant changes are shaded in a dark colour. Suggestive changes (not testing/reaching significance) are shaded in a light colour. Non-significant or significant and conflicting changes are shaded grey.

**Table 1 children-11-00287-t001:** Study characteristics table. Characteristics of included studies. Legend: Time points of summer measures: Pre/post single summer: Changes measured over a single summer period, with data collected at two time points, one at the start and one at the end of summer. Pre/post multiple summers: Study measured changes over more than one summer, with pre-summer and post-summer measures. School comparison: Study compared the outcome between the summer holidays and school period. 1× summer + 1× school: Rather than changes across the summer, the study contrasted an outcome at one school period time point and one summer period time point. Outcomes: Summarizes the outcomes of interest and the type of tool used. Disadvantage: The study considered the impact of disadvantage on the outcomes of interest.

	Participants	Time Points of Summer Measures				Outcomes				
1st Author, Year, Country	Study Setting or Dataset, Participant Age, Sample N = (%Female)	Pre/Post Single Summer	Pre/Post Multiple Summers	School Comparison	1× Summer + 1× School	Academic and Cognitive	Physical Health	Health Behaviours	Mental, Emotional, Social Wellbeing	Disadvantage
Alexander 2007 (USA) [24]	BSS. Age: Grades 1 to Grade 9 N = 790 (51% F).		✓	✓		Reading comprehension (achievement test).				Y
Anderson 2019 (USA) [25]	Extant data collected from one large school district located in the Southwest of the United States (2007–2008 to 2011–2012 school yrs). Age: Grades 3–5 N = 2909 (48% F).		✓	✓		Reading and math (MAP).				Y
Aphamis 2019 (Cyprus) [26]	Students in Limassol, Cyprus. Age: 15–17 yrs N = 153 (44% F).	✓		✓			Adiposity: BMI, % body fat (BIA). Physical fitness: CVF, muscular fitness, flexibility.			N
Atteberry 2021 (USA) [27]	2011–2012 cohort NWEA covering 2008–2016. Age: Grades kindergarten to year eight. N = 260,037 (50% F).		✓	✓		Reading and math (MAP).				N
Benson 2010 (USA) [28]	ELCS-K (fall 1998): Kindergarten to year 1. N = 4180 (% F not reported).	✓		✓		Reading: School achievement and growth rate.				Y
Borman 2006 (USA) [29]	Control data from Teach Baltimore initiative for school students from 10 schools (2 cohorts). Kindergarten and first grade students. Control group data n = 248 (51% F).		✓	✓		Literacy: Reading vocabulary, comprehension, (standardized test).				N
Broekman 2021 (Netherlands) [30]	43 Dutch grade 2 students from schools across five districts. Age: Mean age 7.7 (0.58) range 7–9 yrs. N = 932 (50% F).	✓				Math learning (mental arithmetic speed test).				Y
Brusseau 2018 (USA) [31]	Three low-income elementary schools in a capital city in the Southwestern United States participating in a school-based Comprehensive School Physical Activity Program intervention. Age: Grades 1 to 4. N = 404 (50% F).		✓	✓			Adiposity: BMI Physical fitness: CV fitness (PACER).			N
Brusseau 2019 (USA) [32]	Children in grades 1–5 from traditional and year-round schooling in the same district (Southwestern US). Total sample: n = 321, (traditional 12-week summer break: n = 175, 51% F). Age: Mean age: 9.0 yrs (1.5).	✓					Adiposity: BMI, zBMI Physical fitness: CV fitness (PACER)			N
Burns 2022 (USA) [33]	Children in grades 1 to 4 from 3 low-income schools in the Mountain West region of the United States. Age: 8.9 (1.2) yrs. N = 440 (52% F).	✓					Adiposity: BMI, zBMI.			Y
Campbell 2019 (USA) [34]	Elementary schools from a state in the Southeastern United States. 80% Title 1 schools. Age: Grades 4 and 5. N = 5113 (48% F).	✓		✓		Achievement and learning: Reading (computerized adaptive tests).				N
Chen 2015, 2016 (USA) [35,36]	Students enrolled in kindergarten across 45 elementary schools (Southeast Texas ISD). Age: K to 5th grade. N = 1651 (50% F).		✓	✓			Adiposity: BMI. Weight status: OW/OB > 85%ile OB: >95%ile.			Y
Christodoulos 2006 (Greece) [37]	Greek pre-adolescent elementary school children. Age: 8 (1.4) yrs. N = 183 (53% F).	✓		✓			Adiposity: BMI. Physical Fitness: CVF, muscular strength, power and endurance, agility, flexibility.			Y
Coley 2020 (USA) [38]	ECLS-K Cohort of 2010–2011 (ECLS-K: 2011). Age: 5 yrs, 7 m at baseline. N = approx. 4000 (48% F).		✓	✓		English and math (items from validated and standardized instruments)				Y
Condron 2021 (USA) [16]	NWEA from the Growth Research Database 2002 to 2018. Age: Kindergarten to 10th grade students (11 cohorts N = 10,000,000 (% F not reported).	✓		✓		Skill growth trajectories (MAP).				Y
Daratha 2009 (USA) [39]	Early adolescents in 6th–8th grade from 6 US public middle schools in the Pacific Northwest. Age: 12.25 (0.25) yrs. N = 865 (48% F).	✓		✓			Adiposity: BMI ratio.			Y
Downey 2008 (USA) [40]	ECLS-K: Kindergarten at baseline. N = 4217 (% F not reported).	✓		✓		Reading and math proficiency tests				Y
Downey 2019 (USA) [41]	ECLS Kindergarten cohort 2010–11 Age: Kindergarten to grade 6 students. N = 4029 (% F not reported).		✓	✓					Social and behavioural skills (teacher evaluation questionnaire).	N
Downey 2022 (USA) [42]	NWEA from the Growth Research Database, 6 waves from 2015–2018. Age: 3 cohorts: Kindergarten, third grade and sixth grade at baseline. N = 912,389 (% F not reported).		✓	✓		Reading, math (MAP).				N
Drenowatz 2021 (Austria) [43]	School children from 14 elementary schools in the federal state of Tyrol, Austria. Age: 6.9 (0.4) yrs at baseline N = 214 (48% F).		✓	✓			Adiposity: BMI. Weight status: BMI% (German reference values). Fitness: CV and muscular fitness, flexibility, speed, agility, balance.			Y
Dumont 2019 (USA) [44]	ECLS-K 2010–2011 cohort (ECLS-K:2011): Kindergarten to 2nd grade. N = 16,182. (% F not reported).		✓	✓		Reading and math proficiency.				Y
Economos 2013 (USA) [45]	Public school students in the control group from 2-year RCT conducted in Massachusetts. Control group: Age: Mean 7.5 year. N = 693 (% F not reported).	✓		✓			Adiposity: BMI.			N
Finch 2019 (USA) [46]	ECLS-K: 2010. Nationally representative sample of n = 18,170 children. Age: Not reported. N = 11,150 (% F not reported).		✓	✓		Cognitive function: Working memory (standardized test).				Y
Frisco 2019 (USA) [47]	ECLS-K: 2011 kindergarten, first, and second grade data. Mean age: 5 yrs, 6 months. N = 6500 (48% F).		✓	✓			Adiposity: BMI.			Y
Fryer 2004 (USA) [48]	ECLS-K: Kindergarten and first grade children. N = 13,290 (49% F).		✓	✓		Reading and math (test score gap): Proficiency.				Y
Fu 2017 (USA) [49]	1st to 6th grade children from three low-income schools receiving a Comprehensive School Physical Activity Program (CSPAP) in 2015. Mean age: 9.5 (1.8) yrs. N = 1232 (51% F).	✓					Adiposity: BMI. Health-related fitness: CVF (PACER).			Y
Gershenson 2013 (USA) [50]	Data from two existing time-diary surveys: the Activity Pattern Survey of California Children (APSCC, 1989–1990). Age: range 5–12 yrs. N = 628 (% F not reported).	✓						Time use: Screen (TV), reading, conversation with adults (time-diary interview).		Y
Gorard 2015 (UK) [51]	Control data from experimental study of schools in deprived areas of the UK with 74% ethnic minority. Age: Grade 5 and 6. Total sample n = 196, analytic sample N = 161 (65% F).	✓				Literacy and numeracy (standardized assessments).				Y
Gutin 2008 (USA) [52]	Control data from experimental study of children (59% Black) in grade three at baseline. Mean age: 8.5 (0.6) yrs. Control group data N = 168 (53% F).		✓	✓			Body composition: % body fat (dual-energy x-ray absorptiometry). Adiposity: BMI. Aerobic fitness: Heart rate response to 3 min bench stepping task (heart rate monitor).			N
Helf 2008 (USA) [53]	Control group data from primarily “at-risk, struggling readers” in schools identified as serving low-income families. Age: Kindergarten to grade 2. Control group data: N = 151 (% F not reported).	✓		✓		Literacy (pre and early literacy).				Y
Hunt 2021a (USA) [54]	Control group data from a larger natural experiment on disadvantaged children (59% black, 44.4% low SES). Age: Kindergarten to 4th grade. N = 267 (44% F).	✓						PA: SB, LPA, MVPA and, sleep time (mins, guidelines) (Fitbit Charge 2) and screen: Parent proxy report (questionnaire).		N
Hunt 2021b (USA) [55]	ECLES:K Class of 2010:2011. Mean age: 6.9 (0.8) yrs. N = 1532 (52% F).		✓	✓			Adiposity: BMI. Weight status: 95%ile of CDC age and sex-specific score.			Y
Hwang 2017 (USA) [56]	Control school data from language minority school children from urban Californian schools. Age: Grades 6 to 8. N = 3161 (% F not reported).		✓	✓		Language skills, vocabulary (standardized reading test).				Y
Kim 2004 (USA) [57]	18 ethnically diverse elementary schools in the Lake County Public Schools (LCPS). Age: Grade 5. N = 1627 (% F not reported).	✓		✓		Reading and writing proficiency (pre summer: Citation referenced test, post summer: Achievement test).				N
Kim 2007 (USA) [58]	Suburban public school students. Age: Kindergarten to Grade 6. N = 141 (% F not reported).	✓				Reading achievement and activities (achievement Test).				N
Kim 2008 (USA) [59]	Children in grades 3–5 from two public K-6 elementary schools serving minority populations. Mean age: 8 (range 6–13) yrs. N = 400 (total sample). Controls N = 107. 48% F.	✓				Reading and fluency measures (DIBELS, ITBS).				N
Kim 2010 (USA) [60]	Control group data from experimental study (n = 325) on Latino, language minority students from low-income families who attend Californian public schools. Mean age: 10.3 (SD 5.7 months). Control group data N = 110 (45% F).	✓				Language proficiency. Reading ability (standardized test) (GMRT).				Y
Kobayashi 2006 (Japan) [61]	School Health Statistics in Japan: Data from six waves of elementary school children in Tokyo between 1972 and 2004 (Japanese Ministry of Education). Age: Not reported. N = 446 (52% F).	✓		✓			Body weight (kg). Weight status: “degree of obesity” (comparison to standard values).			N
Kraft 2017 (USA) [62]	Control data from experimental study on public school children in 1st–4th grade, from primarily language minority and low-income families. Mean age: 7.94 yrs. Control group data N = 438 (51% F).	✓		✓		Literacy, reading achievement (standardized test).				N
Kuhfeld 2021 (USA) [63]	NWEA Kindergarten to 8th grade students. Age: Kindergarten to 8th grade (3 cohorts). N = 2,652,382 (% F not reported).		✓	✓		Reading and math (MAP).				Y
Lane 2021 (USA) [64]	Ethnically diverse children from 10 elementary and 2 middle schools from Southwest USA. Age: 8.4 (2.8) yrs (range 6–11). N = 7890 (47.8% F).	✓					Adiposity: BMI, weight status.			Y
Lawrence 2009 (USA) [65]	African American and Hispanic Teens at a mid-sized urban middle school. Age: 6th and 7th grade. N = 192 (% F not reported).	✓				Reading and listening vocabulary and comprehension (GRADE).				Y
Lawrence 2012 (USA) [66]	Urban, mid-sized middle school with extended-day program of serving majority low-income families. Age: 6th and 7th grade. N = 278 (% F not reported).	✓		✓		Vocabulary: Word-learning trajectories (GRADE).				N
Lawrence 2014 (USA) [67]	Control data from experimental study on students at risk for long-term reading difficulty from mostly low-income backgrounds. Age: Grade 7 and 8. (control group data n = 481) (% F not reported).		✓	✓		Vocabulary knowledge (11 target words).				Y
Light 2014 (USA) [68]	Data collected for the “School Social Environments” 3-year longitudinal study of 11 middle schools from western US. Age: Grades 6–8. N = 5742 (52% F).		✓	✓					Antisocial behaviour, social victimisation (youth self-report questionnaires).	N
McClendon 2017 (USA) [69]	Mother–child dyads recruited from “colonias” settlements with informal housing. Age: 11.4 yrs (1.6). N = 121 (52% F).				✓		Adiposity: BMI.	PA: MVPA, SB, screen time (mins), METs (7-day recall instrument).		N
McCoach 2006 (USA) [70]	ECLS-K: Kindergarten and first grade children. N = 8089 (% F not reported).	✓		✓		Reading proficiency (scaled reading assessment).				Y
Meyer 2020 (NZ) [71]	4–7 graders from 60 schools from mid-low socioeconomic areas of New Zealand. Age: 8–12 yrs at baseline. N = 4390 (50% F).	✓		✓		Writing (standardized and norm-referenced test).				Y
Moore 2010 (USA) [72]	Third grade students from 22 classrooms across three rural school corporations with traditional school calendar throughout the state of Indiana. N = 275 (% F not reported).	✓				Math (grade level test).				Y
Moreno 2013, 2015, 2021, 2022 (USA) [73,74,75,76]	Students from 41 elementary schools within a Southeast Texas independent school district from ethnically diverse backgrounds. Mean age: Age range 5–7 yrs (mean 5.7) at baseline. N = 7648 (49% F).		✓	✓			Adiposity: BMI, zBMI. Weight status and trajectory: CDC normative data.	PA: LPA, MVPA, SB (mins). Sleep: mins, timing, variability (wrist-worn accelerometry).		Y
Nagy 2019 (UK) [77]	White British and South Asian children from three schools with a Bradford postcode of various SES levels. Mean age: 7.5 (0.5) yrs, range 6–8). N = 108 (50% F).	✓		✓			Adiposity: BMI, waist circumference.	PA and SB: SB, LP MVPA (mins) (accelerometer).		N
Paechter 2015 (Austria) [78]	Children in grades 5 and 6 from rural Austria. Mean age: 11.1 (0.6) yrs (range 10–12 yrs). N = 110 (55% F).	✓				Math spelling, reading (standardized tests).				N
Patton 2013 (USA) [79]	Elementary school children from schools serving low-income populations (Title 1 schools). Age range: Grades 2–5. N = 317 (55% F).		✓			Reading accuracy and fluency (DIBELS).				Y
Peralta 2022 (Portugal) [80]	Middle school students (6th to 8th grades) from the Lisbon metropolitan area participating in the PA and Family-based Intervention in Paediatric Obesity Prevention in the School Setting (PESSOA) program. Mean age: 12.3 (1.2) at baseline. N = 440 (51% F).	✓		✓			Adiposity: BMI, BMIz. Physical fitness: CVF (PACER).			Y
Quinn 2016, 2018 (USA) [17,81]	ECLS-K: 2011 (ECLS-K:1999 and ECLS-K:2011). Age: Mean 5.7 (SD 4.2 months). N = 14,500 (%F not reported).		✓			Reading and math scores and inequality gaps (NAEPF).				Y
Rambo-Hernandez 2015 (USA) [82]	Grade 3–6 students. Age: Grade 3 at baseline. Analytic sample: N = 70,521 (49% F).		✓	✓		Reading and comprehension: Growth rate (MAP).				N
Reed 2020, 2021 (USA) [83,84]	Archival data on students from 43 school districts. Age: Kindergarten to year 5. N = 9971 (% F not reported).		✓			Reading ability and progress (various grade level proficiency, achievement test).				Y
Reesor 2019 (USA) [85]	Secondary analysis of data from five experimental studies (n = 425) on low income, Hispanic middle school students from grades 6 and 7 (2005–2010). Mean age: 12.08 (0.63). Control group data N = 230 (50% F).	✓		✓			Adiposity: BMI, zBMI. Weight status.			N
Rodriguez 2014 (USA) [86]	Low-income Hispanic elementary school children in 3rd and fourth grades from metropolitan schools in Greater Houston (Texas). Age: 9.2 (0.8) yrs. N = 119 (58% F).	✓		✓			Adiposity: % body fat, BMI, weight status. Fitness: Aerobic endurance (PACER).			N
Rulison 2010 (USA) [87]	Primarily Caucasian students from a single elementary school serving a rural, working-class community. Age: Grades 3–5. N = 427 (45% F).		✓	✓					Self-worth, self-perception (Self-Perception Profile for Children).	N
Sallis 2019 (USA) [88]	Adolescents from lower-income areas of five states (2017–2018). Mean age: 13.9 (SD 1.85) yrs, range 10–17 yrs. N = 207 (47% F).				✓			PA: MVPA, SB (mins/day) (belt-worn Actigraph accelerometer), screen time (mins) (survey).	Psychosocial measures of PA (self-efficacy, social support).	N
Schaffner 2016 (Germany) [89]	3rd grade elementary students from different SES backgrounds from 15 federal schools in urban and rural areas. Mean age: 8.9 (0.59) yrs. N = 223 (45% F).	✓				Reading amount, frequency, motivation (questionnaire), comprehension (subtests from standardized test).				Y
Skibbe 2012 (USA) [90]	Children from 16 schools from one school district in a suburban Midwest town. Age (mean): K: (5.4 yrs), grade 1 (6.5 yrs), grade 2 (7.4 yrs). N = 288. (51.4% F).		✓	✓		Literacy growth. Children’s literacy (achievement, vocabulary and comprehension tests).				N
Slates 2012 (USA) [91]	Subset of BSS from 1982: Low SES elementary school children who exhibit high summer achievement over 4 summers. Age at baseline: Not reported. N = 790 (% F not reported).		✓	✓		Reading and math achievement (CAT).				N
Smith 2009 (USA) [92]	American Indian Children 7–14 years old at baseline (Summer sub-sample). Mean age: not reported. N = 141 (49% F).	✓					Adiposity: BMI, zBMI. Weight status.			N
Takacs 2020 (Hungary) [93]	Control data from children (8 classes, 2 schools) of an experimental nutrition study. Whole sample mean age: Grades 6 and 7, mean 12.6 (SD 0.1) yrs. Control group data only (N = 122) (55% F).	✓		✓			Adiposity: Waist circumference, BMI. Fitness (NETFIT protocol): CVF, muscular strength, flexibility.	Diet (breakfast, meal number and frequency, food categories) (questionnaire).		N
Tanaka 2016, 2018 (Japan) [94,95]	Japanese primary children from 4 public primary schools in urban areas in Tokyo and Kyoto. Age: 9.0 (1.8) yrs at baseline. N = 209 (53% F).	✓					Adiposity: BMI. Weight status: Relative body weight (Japanese age and sex reference data).	PA: SB, LPA, MVPA, MPA and VPA, steps/day (waist-worn triaxial accelerometer), screen time, diet (questionnaire).		Y
Tanskey 2019 (USA) [96]	Control data from experimental study of 3rd and 4th grade children participating in a school-based PA program evaluation in urban, racially diverse, public elementary schools. Mean age: 9.18 (SD 0.64) yrs. Control group data N = 769 (56% F).	✓		✓			Adiposity: BMI. Weight status: CDC growth percentiles.	PA: SB, LPA, MPA, VPA (mins/day) (waist-worn accelerometer). Diet (FLEX Dietary questionnaire).		N
Tiruchittampalam 2018 (New Zealand) [97]	Five year holds attending higher or lower SES schools. Age: Range 5–5.3 yrs. N = 126 (56% F).		✓	✓		Pre-literacy knowledge (various measures across time points).				Y
vonHippel 2016 (USA) [9]	ECLS-K: Kindergarten Class of 2010–11. Age range: Kindergarten to 2nd grade N = 18,170 (% F not reported).		✓	✓			Adiposity: BMI. Weight status (overweight and obesity prevalence).			Y
vonHippel 2018 (USA) [98]	ECLS-K: 1999 and 2011 cohorts. Age range: Kindergarten to 2nd grade students N = 34,630 (% F not reported).		✓			Reading and math learning rates and inequity gaps (ability scores and scale scores).				N
Weaver 2020a [99], 2020b [100], 2021 [101] (USA)	Elementary-aged school children attending year-round or traditional schools. Primarily low-income households with majority Black students. Age range: 5–12 yrs. N = 2279 (51%F). Behavioural data subset N = 240 (51% F).		✓	✓			Adiposity: BMI, zBMI, weight status. Fitness: CVF (PACER).	PA and sleep: SB, LPA, MVPA (mins), sleep (mins, mid-point, efficiency) (Fitbit Charge 2). Time use: Screen time, diet (parent proxy-report, questionnaire).		Y
White 2014 (USA) [102]	Control data from students from a midsize urban school district enrolled in an experimental study. Age: Grade 3. Control group data N = 792 (% F not reported).	✓				Reading comprehension (ITBS tests).				N
Yoon 2018 (USA) [103]	NWEA data on white and Asian children from approx. 675 public schools in the US. Age: Grades K to 7 N = Approx 130,000 (% F not reported).		✓	✓		Reading and math (MAP).				N
Zhang 2011 (USA) [104]	Control data from experimental study of American Indian Children from the Pine Ridge Reservation enrolled in the Bright Start RCT school-based intervention to reduce weight gain. Age: 5.8 (0.5), range 4.8–7.9 yrs. N = 454 (48% F).		✓	✓			Adiposity: BMI. Weight status: growth velocity (CDC growth charts).			N
Zvoch 2011 (USA) [105]	Administrative records of a moderately sized school district in the Pacific Northwest (41% low-income families). Mean age: 6.4 (0.25) yrs. N = 1449 (48% F).	✓				Reading (test of oral reading fluency).				N

Abbreviations: ✓ denotes study design includes these features. BIA: bioelectric impedance analysis, BSS: Baltimore-based Beginning School Study, CAT: California Achievement Test. CDC: Centers for Disease Control, CVF: Cardiovascular fitness, DIBELS: Dynamic Indicators of Basic Early Literacy Skills Oral Reading Fluency, ECLS: Early Childhood Longitudinal Study, GRADE: Group Reading Assessment and Diagnostic Evaluation test, ISD: Independent school district, ITBS: Iowa Tests of Basic Skills, K: Kindergarten, MAP: Measures of academic progress, METs: Metabolic equivalent of task, MVPA: Moderate–vigorous physical activity, NAEPF: National Assessment of Educational Progress Framework, NWEA: Northwest Evaluation Association data set, OB: Obese, OW: Overweight, PA: Physical activity, PACER: 20-metre progressive aerobic cardiovascular endurance run, SB: Sedentary behaviour, Yrs: years.

## Data Availability

The data presented in this study are available in article and Appendix A.

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
