# Peer review of "Children’s Health, Wellbeing and Academic Outcomes over the Summer Holidays: A Scoping Review"

_children, 2024, doi:10.3390/children11030287_

Round 1
Reviewer 1 Report
Comments and Suggestions for Authors
Overall this is a well written paper contributing to the area in question. It has been written well and in terms of the scoping review has been conducted well. There are some amendments required to be made:
Abstract
Line 43 – what is meant be disadvantaged? Expand/make clearer here to support what you are actually referring to.
Introduction
Line 76 – ‘heath over the summer’.
Methods
Lines 140-141 – dissolved by consensus, is between the first author and one other, or all second authors please provide the detail.
Line 142 – who conducted the reference list search one, all or some authors – please specify.
Lines 170-171 put ‘years’ after the ages as unit of measure.
Results
Line 211- 12 years.
Lines 212 and 213 – add ‘years’ please.
Line 213 – through to five
Do the Atteberry, Benson, Borman, Brusseau, Campbell, Chen, Condron, Downey, x 3, Dumont, Fryer, Gorard, Helf, Hunt, Hwang, Kim x2 Kuhfeld, Lawrence x 3,Light, McCoach, Moore, Patton, Rambo, Reed, Rulison, Tackacs, vonHippel, white, Yoon, study state actual age? Please add in ages so readers for all countries can clearly see the age when they look through table 1.
End of line 19-20 (page 23) – doesn’t flow possibly remove ‘of’ before ‘academic’.
Figure 3, 4, 5 and 6 a key showing the darker and lighter colours would be advisable.
Figure 3 – I can not see the title for the y-axis?
Line 48 – Figure 4?
Lines 51-54 – the number of studies stated on lines 52 to 54 equals 22 NOT 23 as stated on line 51.
Lines 57-60 - when broken down the figure totals 9 NOT the 10 stated on line 57.
Line 75 – over the summer.
Line 77 – remove ‘study’ from before Japanese.
Lines 102-105 – state the actual gap/direction of gap between high and low SES countries.
Line 113 – put the comma next to populations.
Discussion
Implications section – who would actually provide the summer programmes, what local arrangements could be implemented, what cost implications would be involved and to who – policy makers or families. More within this section and what you see as future developments would be advantageous.
Supplementary 1 – Inclusion criteria for population – repetition of ‘of the’ – please remove.
Supplementary 3 – Physical Health adds up to 11 not 10?
Comments on the Quality of English LanguageThe English Language within the paper was to a high standard, minimal errors have been identified.
Author Response
Thank you for your review. Please see the attached document.

Reviewer 2 Report
Comments and Suggestions for Authors
Rewiew
Children’s Health, Wellbeing and Academic Outcomes over the 2 Summer Holidays: A Scoping Review
The research on "Children’s Health, Wellbeing and Academic Outcomes over the 2 Summer Holidays" is crucial due to its focus on significant health and educational trends, such as the rise in childhood obesity and the academic "summer slide." It emphasizes the need for targeted interventions during summer breaks, especially as these periods exacerbate disparities for disadvantaged children. Highlighting over 14 million participants, the study underscores the broad impact of summer holidays on children's health and academic performance, advocating for the development of programs to counteract sedentary behaviors and learning losses. Its timely insights offer guidance for policymakers and educators, pointing to areas needing further research and action to support child wellbeing and equity in education. In summary, this research is relevant and up-to-date because it addresses significant current health and educational concerns, highlights the amplified impact on disadvantaged children, suggests opportunities for positive interventions, and guides future research priorities.
The main criticism identified in the scientific article's conclusion is the lack of data on children's social, emotional, and mental wellbeing changes across the summer. This gap signifies a critical area of research that has not been adequately addressed, leaving a partial understanding of how the summer holidays affect children's overall health and wellbeing. The conclusion suggests that while there is strong evidence of negative physical health and academic outcomes during the summer, especially for disadvantaged children, the impact on the broader aspects of children's wellbeing remains unclear due to insufficient evidence. This limitation underscores the need for further research to fully understand the scope of summertime changes in children's lives and to develop comprehensive interventions. Write about it in the limitation.
The methodology is good, perhaps the text on the figures could be in a larger font size so that it can be seen better.
Author Response
Thank you for taking the time to review this manuscript. Please see the attached document.
